# Negative regulation of miRNA sorting into EVs is mediated by the capacity of RBP PCBP2 to impair the SYNCRIP-dependent miRNA loading

Francesco Marocco[1†], Sabrina Garbo[1†], Claudia Montaldo[2], Alessio Colantoni[3,4], Luca Quattrocchi[1], Gioele Gaboardi[1], Giovanna Sabarese[5], Carla Cicchini[1], Mario Lecce[1], Alessia Carnevale[1], Rossella Paolini[1], Gian Gaetano Tartaglia[6], Cecilia Battistelli[1*†], Marco Tripodi[1,2*†]

[1]Istituto Pasteur Italia-Fondazione Cenci Bolognetti, Department of Molecular Medicine, Department of Excellence 2023-2027, Sapienza University of Rome, Rome, Italy; [2]National Institute for Infectious Diseases L. Spallanzani, IRCCS, Rome, Italy; [3]Biology and Biotechnology Department C. Darwin, Sapienza University of Rome, Rome, Italy; [4]Center for Life Nano- and Neuro-Science, RNA Systems Biology Lab, Fondazione Istituto Italiano di Tecnologia (IIT), Genoa, Italy; [5]Anatomical Pathology Operative Research Unit, Fondazione Policlinico Universitario Campus Bio-Medico, Rome, Italy; [6]Center for Human Technologies, Istituto Italiano di Tecnologia, Genoa, Italy

*For correspondence:
cecilia.battistelli@uniroma1.it (CB);
marco.tripodi@uniroma1.it (MT)

†These authors contributed equally to this work

Competing interest: The authors declare that no competing interests exist.

## eLife Assessment

This article makes a **valuable** contribution to the field by uncovering a molecular mechanism for miRNA intracellular retention, mediated by the interaction of PCBP2, SYNCRIP, and specific miRNA motifs. The findings are **convincing** and advance our understanding of RNA-binding protein-mediated miRNA sorting, providing deeper insights into miRNA dynamics.

**Abstract** While it is accepted that extracellular vesicles (EVs)-mediated transfer of microRNAs contributes to intercellular communication, the knowledge about molecular mechanisms controlling the selective and dynamic miRNA-loading in EVs is still limited to few specific RNA-binding proteins interacting with sequence determinants. Moreover, although mutagenesis analysis demonstrated the presence/function of specific intracellular retention motifs, the interacting protein/s remained unknown. Here, PCBP2 was identified as a direct interactor of an intracellular retention motif: CLIP coupled to RNA pull-down and proteomic analysis demonstrated that it binds to miRNAs embedding this motif and mutagenesis proved the binding specificity. Notably, PCBP2 binding requires SYNCRIP, a previously characterized miRNA EV-loader as indicated by SYNCRIP knock-down. SYNCRIP and PCBP2 may contemporarily bind to miRNAs as demonstrated by EMSA assays and PCBP2 knock-down causes EV loading of intracellular microRNAs. This evidence highlights that multiple proteins/miRNA interactions govern miRNA compartmentalization and identifies PCBP2 as a dominant inhibitor of SYNCRIP function in murine hepatocytes.

## Introduction

Extracellular vesicles (EVs) budding from cellular plasma membrane and differing for size and origin (e.g., exosomes, of 30–150 nm diameter and microvesicles, which diameter is ≥50 nm) (*van Niel et al., 2022*) are uptaken by neighboring as well as by distant recipient cells, thus transferring their specific informational cargo of molecules. Indeed, EVs are now considered a well-assessed route for donor cells to promote changes in gene expression and cell behavior of recipient cells in several pathophysiological processes (*Mathieu et al., 2019*; *Montaldo et al., 2021*). Notably, concerning miRNAs, the EV-cargo does not simply reflect the cell of origin content, but rather is defined by dynamic and selective cell-specific loading mechanisms (*Garbo et al., 2024*; *Garcia-Martin et al., 2022*; *Mori et al., 2019*; *Santangelo et al., 2016*; *Villarroya-Beltri et al., 2013*). The existence of multiple players controlling miRNA compartmentalization matches with the hypothesis of a multiprotein machinery that dynamically governs not only the EV sorting but also, conceivably, the intracellular retention of specific subsets of miRNAs.

At present, the molecular players controlling the selective partition of miRNAs seem largely uncharacterized, although recent evidence highlighted the mechanisms of EV loading depending on sequence-specific RNA-binding proteins (RBPs). In this regard, the first identified was the Heterogeneous Nuclear Ribonucleoprotein A2B1 (hnRNPA2B1), recognizing miRNAs containing the EXO motif (G/U,G,A/G/C,G/C) and guiding their inclusion in EVs secreted by human primary T-lymphocytes (*Villarroya-Beltri et al., 2013*). A functional role was successively attributed to the Synaptotagmin-binding Cytoplasmic RNA-Interacting Protein (SYNCRIP or hnRNPQ) that, by binding to miRNAs embedding a short hEXO motif (G/A/U,G/U/A,G/A/U,C/A/G,U/A,G/C) through its NURR domain and its RRMs recognizing the 5' of the motif (*Santangelo et al., 2016*; *Hobor et al., 2018*), guides their inclusion in EVs secreted by hepatocytes. Notably, hEXO was shown to increase the EV/cell ratio of miRNAs as demonstrated by the insertion of this sequence in a cell-retained miRNA (*Santangelo et al., 2016*; *Hobor et al., 2018*), thus paving the way for further manipulative perspectives in the growing field of RNA-based therapies (*Garbo et al., 2022*; *Paunovska et al., 2022*; *Traber and Yu, 2023*).

More recently, similar functional evidence gathered by the chimeric approach has been extended to newly identified motifs, causing miRNAs export or intracellular retention (*Garcia-Martin et al., 2022*). While for these export motifs two RNA-binding proteins (ALYREF and FUS) have been characterized (*Garcia-Martin et al., 2022*) for the retention motifs, defined as 'CELL' (*Garcia-Martin et al., 2022*), as for the previously described 'CL' motifs (*Villarroya-Beltri et al., 2013*) the identification of interacting protein/s remains unaddressed.

We here aimed at the investigation of molecular players responsible for miRNAs intracellular retention. Overall, we gathered evidence on the interactions among several CELL-embedding miRNAs and two RBPs: SYNCRIP and the multifunctional RNA-binding protein PCBP2. Functionally, PCBP2 promotes cellular retention in SYNCRIP-bound miRNAs and the already described SYNCRIP loading activity appears impeded by PCBP2.

## Results

### PCBP2 recognizes a CELL motif and has a functional role in the intracellular retention of miRNA-155-3p

Aiming to identify the RBPs involved in the intracellular retention, proteins from hepatocytes were used in RNA-pull-down by using as specific bait the miRNA-155-3p, selected for the presence of the Core CELL-motif identified in AML12 cells (*Garcia-Martin et al., 2022*; *Figure 1A*).

Label-free nLC-MS/MS proteomic analysis allowed to identify miR-155-3p-interacting proteins (*Supplementary file 1*), and label-free quantification intensities analysis identified 21 proteins enriched in miR-155-3p pull-down with respect to miR-155-3p mutated in the Core CELL motif (miR-155-3p no-CELL) (*Figure 1A*, *Supplementary file 2*, and *Figure 1B*). Twelve of them are classified as RNA-binding proteins, with six containing at least one canonical RNA-binding domain (*Cook et al., 2011*; *Liao et al., 2020*; *Ray et al., 2013*; *Treiber et al., 2017*, *Supplementary file 2*). For three of them, RNA-binding preferences are also known (i.e. PCBP2 prefers CU-rich sequences; *Ray et al., 2013*; *Van Nostrand et al., 2020*), LARP1 recognizes the CAP and the 5' top motif in mRNAs (*Fonseca et al., 2015*), and STAU1 binds to double-stranded RNAs (*Wickham et al., 1999*).

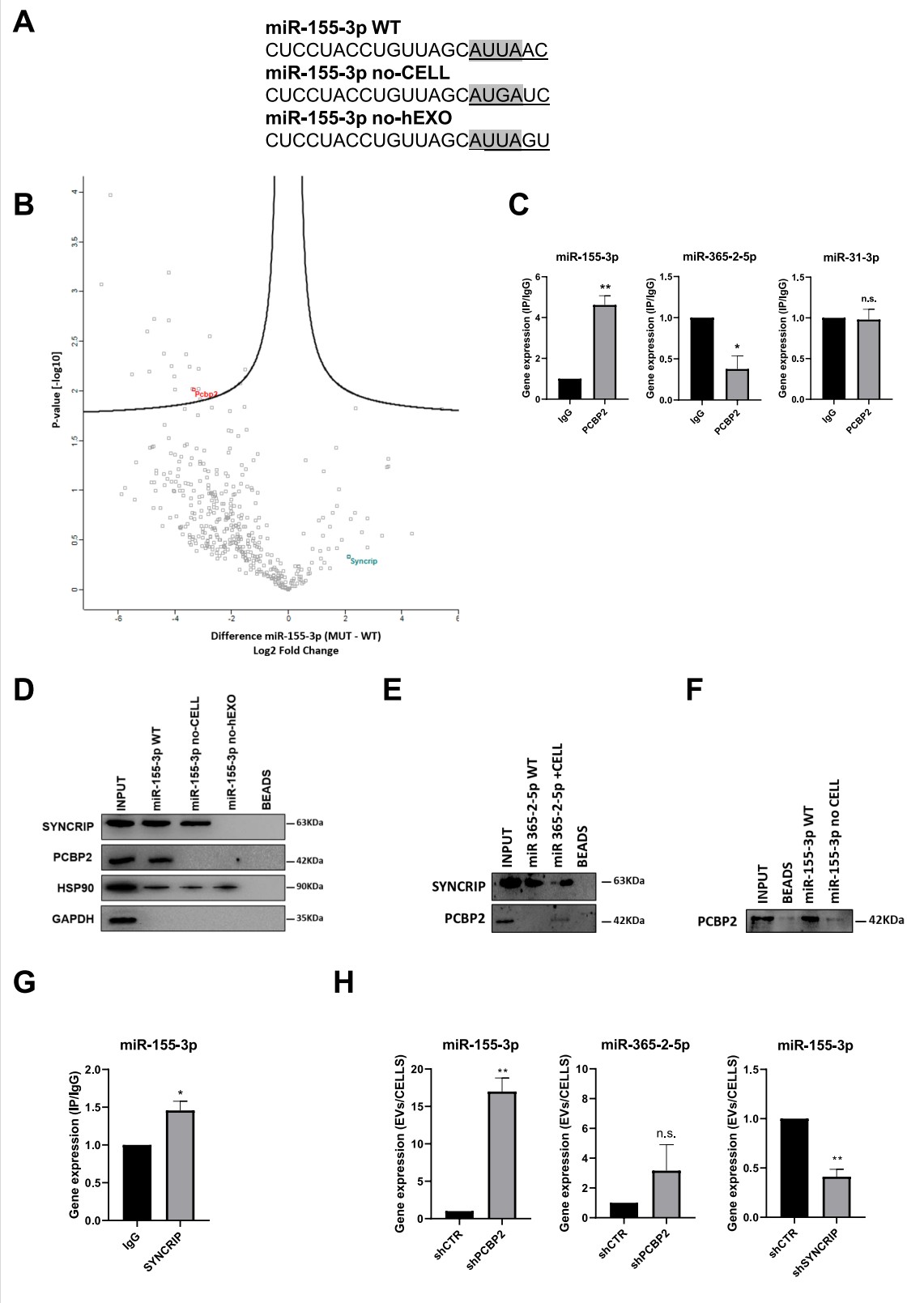

**Figure 1.** PCBP2 recognizes the CELL motifs and has a functional role in intracellular retention of miRNA-155-3p. (**A**) Sequences of biotinylated oligos used as bait in pull-down experiment; Core CELL motifs (WT and mutated) are in gray, hEXO motifs (WT and mutated) are underlined. miRNA devoid of CELL motif (no-CELL), miRNA devoid of hEXO (no-hEXO). (**B**) Volcano plot comparing proteins bound to miRNA-155-3p no-CELL vs WT. Black curves represent the significant threshold at a false discovery rate (FDR) of 0.05 and S0 of 0.1. PCBP2 and SYNCRIP proteins are labeled in the plot. (**C**) CLIP of

*Figure 1 continued on next page*

*Figure 1 continued*

PCBP2 protein in murine hepatocytes. RT-qPCR analysis for miR-155-3p, miR-365-2-5p (CELL motif-devoid), and miR-31-3p (hEXO motif-devoid) is shown as IP/IgG. Data are the mean ± SEM of three independent experiments. (**D**) RNA pull-down with the WT and mutated (no-CELL, no-hEXO) (sequences are reported in **A**) miR-155-3p followed by western blot for the indicated proteins (HSP90 is used as positive and GAPDH as negative controls, respectively). Data are representative of three independent experiments. (**E**) RNA pull-down with the WT and mutated (+CELL) miR-365-2-5p followed by western blot for the indicated proteins. Data are representative of three independent experiments. (**F**) RNA pull-down by using the recombinant PCBP2 protein and with WT and mutated miR-155-3p (no-CELL) followed by western blot for PCBP2. Data are representative of three independent experiments. (**G**) CLIP of SYNCRIP protein in murine hepatocytes. RT-qPCR analysis for miR-155-3p is shown as IP/IgG. Data are the mean ± SEM of three independent experiments. (**H**) (Left and middle panels) EV miRNA-155-3p and miR-365-2-5p levels in *shCTR* and *shPCBP2* cells analyzed by RT-qPCR. Data are expressed as ratio of miRNA expression in EVs with respect to the intracellular compartment (*shCTR* arbitrary value 1). Results are shown as the mean ± SEM of three independent experiments. (right panel) EV miRNA-155-3p levels in *shCTR* and *shSYNCRIP* cells analyzed by RT-qPCR. Data are expressed as ratio of miRNA expression in EVs with respect to the intracellular compartment (*shCTR* arbitrary value 1). Results are shown as the mean ± SEM of three independent experiments. Data are considered statistically significant with p<0.05 (Student's *t*-test). *p<0.05; **p<0.01.

The online version of this article includes the following source data and figure supplement(s) for figure 1:

**Source data 1.** Original western blots for *Figure 1D–F*, indicating the relevant bands.

**Source data 2.** Original files for western blots analysis displayed in *Figure 1D–F*.

**Figure supplement 1.** PCBP2 and SYNCRIP silencing.

**Figure supplement 1—source data 1.** Original western blots for *Figure 1—figure supplement 1B and D* indicating the relevant bands.

**Figure supplement 1—source data 2.** Original files for westerns blots displayed in *Figure 1—figure supplement 1B and D*.

**Figure supplement 2.** *shCTR, shPCBP2,* and *shSYNCRIP* EV characterization.

**Figure supplement 2—source data 1.** Original western blots for *Figure 1—figure supplement 2A*, indicating the relevant bands.

**Figure supplement 2—source data 2.** Original files for western blots analysis displayed in *Figure 1—figure supplement 2A*.

Among them, PCBP2 interaction with miR-155-3p was confirmed by CLIP assay (*Figure 1C*, left panel) and RNA pull-down followed by western blot analysis (*Figure 1D*). A second proof of concept of the requirement of PCBP2/CELL motif interaction is provided by the observation on the CELL motif-devoid miR-365-2-5p (*Figure 1C*, middle panel); RNA pull-down confirmed the absence of interaction with PCBP2 (*Figure 1E*). Notably, the inclusion of the Core CELL motif while maintaining SYNCRIP binding, allows PCBP2 interaction with this miRNA. At least in vitro, this binding appears direct and sequence specific as demonstrated using a recombinant protein in RNA pull-down (*Figure 1F*). The introduction of specific mutations allowed to test the requirement of the CELL motif since its modification (miR-155-3p no-CELL) impairs PCBP2 binding (*Figure 1D*).

Notably, the inspection of the miR-155-3p sequence revealed the presence of the previously identified SYNCRIP binding site and both MS/MS analysis (*Supplementary file 1*) and CLIP assay (*Figure 1G*) confirmed SYNCRIP binding to this miRNA independently of the CELL motif mutation (*Figure 1D*).

Surprisingly, the permutation of the downstream two nucleotides removing SYNCRIP-binding motif (miR-155-3p no-hEXO, *Figure 1A*) impairs PCBP2 binding despite the conservation of the CELL retention motif (*Figure 1D*).

This suggests a possible SYNCRIP requirement for PCBP2 binding. To test this hypothesis, CLIP assay was performed on miR-31-3p, embedding the sole CELL motif, indicating the absence of PCBP2 binding (*Figure 1C*, right panel).

Functionally, the role of PCBP2 in miRNA partition was addressed by its silencing (*Figure 1—figure supplement 1A and B*). Notably, PCBP2 interference (PCBP2 being not expressed in the EVs, *Figure 1—figure supplement 2A*) enhances miRNA-155-3p loading in EVs with respect to control cell-derived EVs (*Figure 1H*, left panel). As a further control that miRNAs without the CELL motif are not affected by PCBP2 silencing, the expression levels of miR-365-2-5p (embedding the sole hEXO motif) were analyzed in EVs and cells, and resulted not differentially exported (*Figure 1H*, middle panel). As expected, SYNCRIP silencing (*Figure 1—figure supplement 1C and D*) (expressed in both intracellular and EV compartment as shown in *Figure 1—figure supplement 2A*) reduces miR-155-3p export into EVs (*Figure 1H*, right panel; for EVs characterization, see *Figure 1—figure supplement 2A and B*). Moreover, the effect of either PCBP2 or SYNCRIP knockdown on EVs production was ruled out by quantifying the number of EVs per cell in *shPCBP2* or *shSYNCRIP* with respect to the *shCTR* conditions (*Figure 1—figure supplement 2C*).

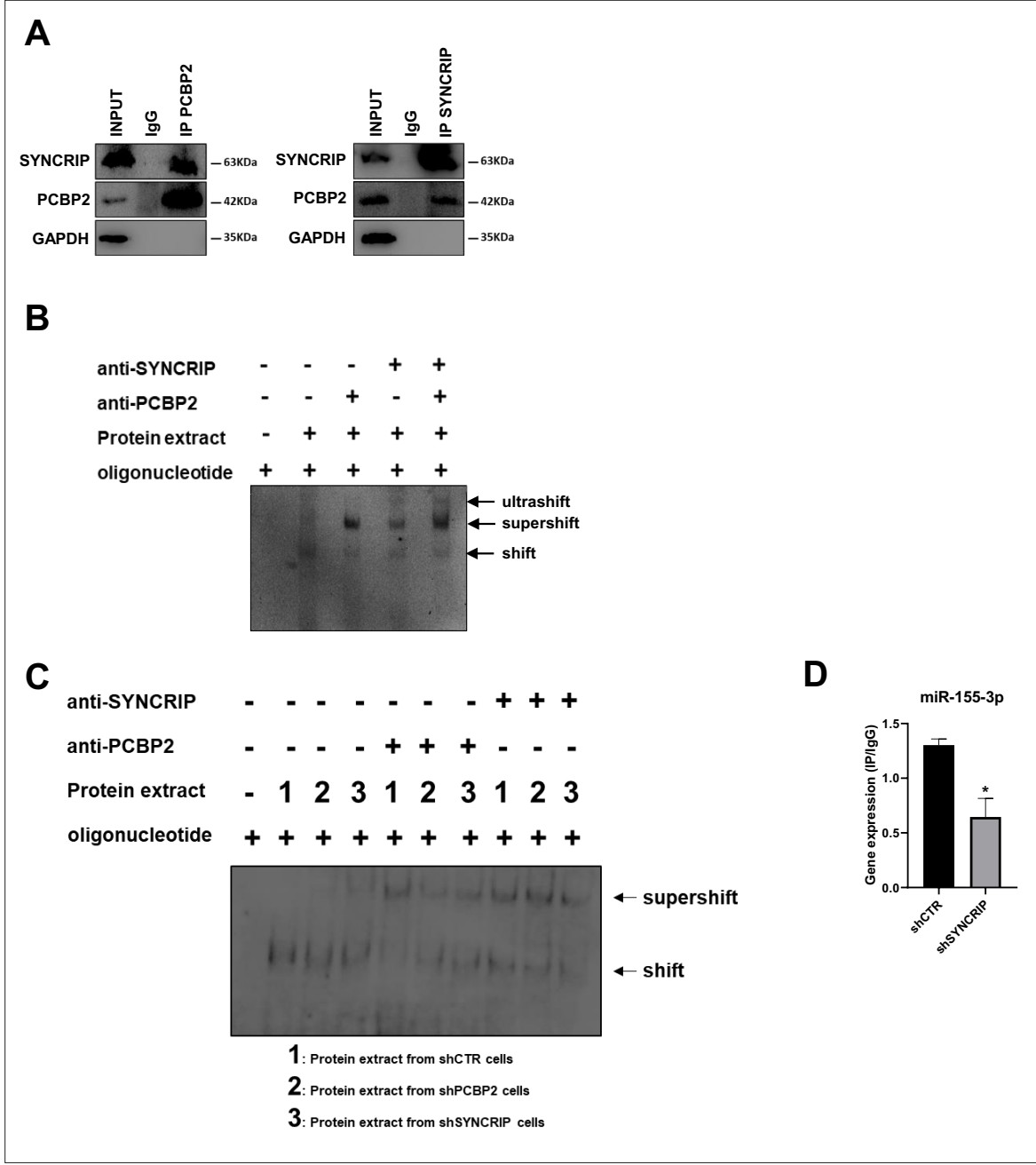

**Figure 2.** PCBP2 binding to miR-155-3p is SYNCRIP-dependent. (**A**) Co-immunoprecipitation of PCBP2 and SYNCRIP. Immunoprecipitations with rabbit polyclonal anti-PCBP2, mouse monoclonal anti-SYNCRIP, and the relative preimmune IgG were performed on protein extracts from hepatocytes. GAPDH is used as negative control. Immunoblots representative of three independent experiments are shown. (**B**) Electrophoretic mobility shift assay (EMSA): interactions of miR-155-3p with the indicated protein extracts (shifts) and Abs (anti-SYNCRIP and anti-PCBP2) (supershift) are shown. Ultrashift shown in lane 5 demonstrates concurrent binding of SYNCRIP and PCBP2 to miR-155-3p. (**C**) EMSA: interactions of miR-155-3p with protein extracts from *shCTR* (1), *shPCBP2* (2), and *shSYNCRIP* (3) cells (shifts) and Abs (anti-SYNCRIP and anti-PCBP2) (supershift) are shown. (**D**) CLIP of PCBP2 protein in murine hepatocytes both WT (*shCTR*) and silenced for SYNCRIP (*shSYNCRIP*). RT-qPCR analysis for the expression of miR-155-3p is shown as IP/IgG. Data are the mean ± SEM of three independent experiments.

The online version of this article includes the following source data for figure 2:

**Source data 1.** Original western blots and EMSA analysis for *Figure 2A– C*, indicating the relevant bands.

**Source data 2.** Original files for western blots and EMSA analysis displayed in *Figure 2A–C*.

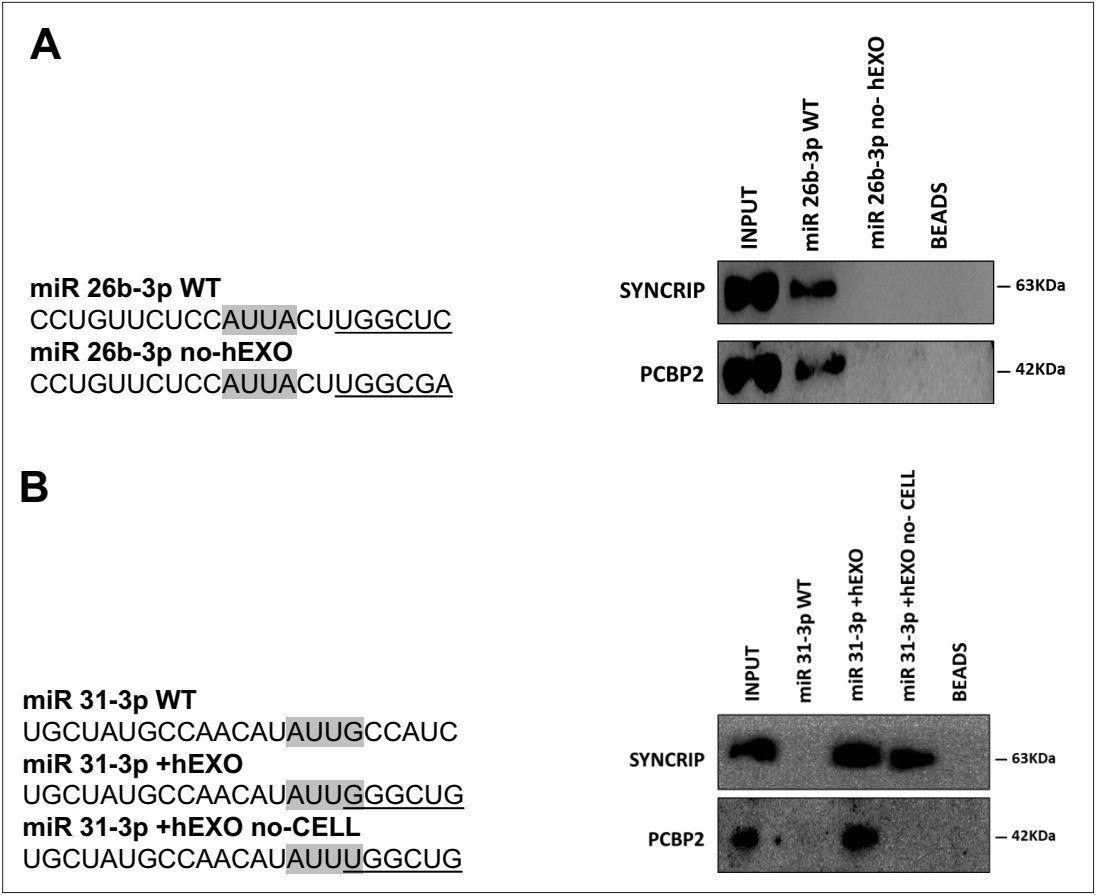

**Figure 3.** PCBP2 binding to miR-155-3p is sequence dependent. (**A**) RNA pull-down with the WT and mutated (sequences are reported above) miR-26b-3p followed by western blot for the indicated proteins. Data are representative of three independent experiments. (**B**) RNA pull-down with the WT and mutated (sequences are reported above) miR-31-3p followed by western blot for the indicated proteins. Data are representative of three independent experiments. (**A**, **B**) Core CELL motifs (WT and mutated) are in gray, hEXO motifs (WT and mutated) are underlined.

The online version of this article includes the following source data for figure 3:

**Source data 1.** Original western blots for **Figure 3A and B**, indicating the relevant bands.

**Source data 2.** Original files for western blots displayed in **Figure 3A and B**.

Overall, these data demonstrated that (i) PCBP2 interacts with miRNA-155-3p, as proved by CLIP analysis and RNA pull-down, (ii) the interaction is CELL-motif-dependent while an unexpected role for the hEXO motif is also unveiled, and (iii) PCBP2 favors the intracellular localization of this miRNA.

## PCBP2 binding to miR-155-3p is both sequence- and SYNCRIP-dependent

The observation that loading (hEXO) and retention (CELL) motifs are both present in miR-155-3p sequence prompted us to investigate on the hypothesis of a sequence- and SYNCRIP-dependent PCBP2 binding ability.

First, we observed that the two proteins interact each other (**Figure 2A**) and more interestingly that both RBPs bind to miR-155-3p contemporarily as indicated by the ultra-shift obtained in EMSA assay (**Figure 2B**).

To challenge the hypothesis of a SYNCRIP-dependent PCBP2 binding, EMSA assay was performed in PCBP2-silenced and in SYNCRIP-silenced cells (**Figure 1—figure supplement 1A–D** respectively). As shown in **Figure 2C** while PCBP2 silencing does not affect SYNCRIP binding, SYNCRIP silencing impairs also PCBP2 binding. Furthermore, CLIP assay was performed on SYNCRIP-silenced cells; as shown in **Figure 2D**, SYNCRIP silencing impairs PCBP2 binding to miR-155-3p.

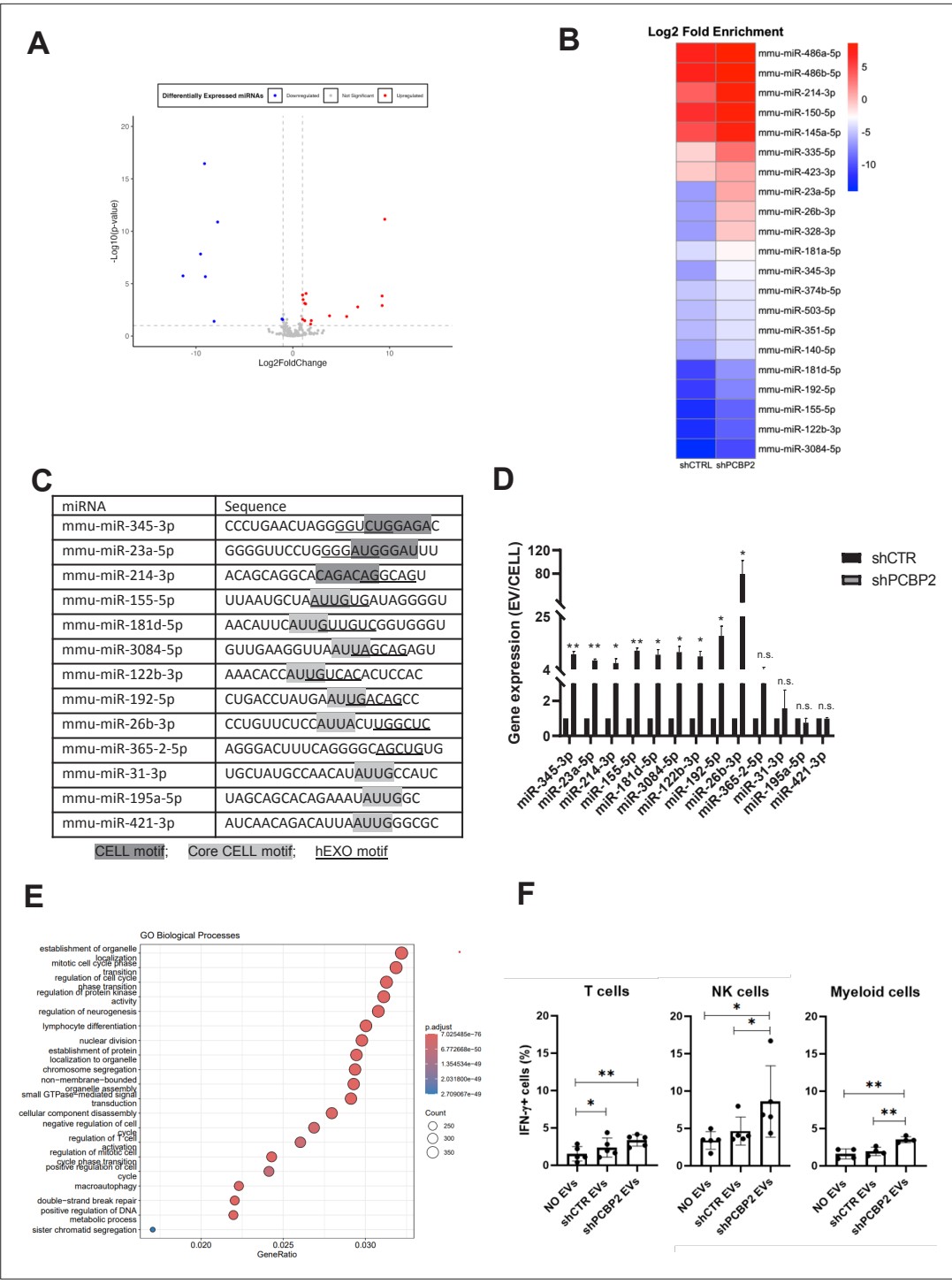

**Figure 4.** PCBP2 functionally dominates on SYNCRIP EV-loading activity on a repertoire of miRNAs embedding CELL and hEXO motifs. (**A**) Volcano plot comparing miRNAs differentially expressed from NGS data; miRNAs with Log2FC > 1 and Log2FC < -1 and p-value ≤ 0.10 were considered differentially expressed. Downregulated miRNAs in *shPCBP2* respect to *shCTRL* are represented as blue dots, upregulated miRNAs are represented as red dots. (**B**) Heatmap showing the Log2 fold enrichment (EV/CELL) of mature miRNAs in small extracellular vesicles derived from *shCTRL* cells versus *shPCBP2* cells. miRNAs with Log2FE ≥ 1.0 and p-value ≤ 0.10 were considered to be differentially enriched. (**C**) List of selected miRNAs embedding CELL and/or hEXO motifs; consensus sequences are highlighted in gray or underlined respectively. (**D**) EV miRNA levels in *shCTR* and *shPCBP2* cells analyzed by RT-qPCR. Data are expressed as ratio of miRNA expression in EVs with respect to the intracellular compartment

*Figure 4 continued on next page*

*Figure 4 continued*

(*shCTR* arbitrary value 1). Results are shown as the mean ± SEM of three independent experiments. Data are considered statistically significant with p<0.05 (Student's *t*-test). *p<0.05; **p<0.01. (**E**) Gene Ontology enrichment analysis on validated and predicted targets. The X-axis represents the gene ratio, whereas the Y-axis represents the enriched GO terms. Colors indicate the statistical significance after multiple test correction, while circle size represents the number of genes associated with each term. (**F**) Liver-isolated leukocytes cultured either alone (NO EVs) or in the presence of control EVs (*shCTR* EVs) or *shPCBP2* EVs (*shPCBP2* EVs) for 24 h. The expression of IFN-γ was analyzed on CD3$^+$, NK1.1$^+$ CD11b$^+$, and CD11b$^+$ cells by flow cytometry. The dot plots show the percentage of cytokine-positive T cells, NK cells, and myeloid cells. T cells and NK cells graphs are representative of five independent experiments, while myeloid cells graph is representative of four independent experiments. Each symbol represents data obtained from an individual mouse. Data are considered statistically significant with p<0.05 (Student's *t*-test). *p<0.05; **p<0.01.

The online version of this article includes the following figure supplement(s) for figure 4:

**Figure supplement 1.** miRNAs EV export upon SYNCRIP silencing.

**Figure supplement 2.** Gene Ontology analysis on miRNA targets.

**Figure supplement 3.** Flow cytometric analysis on EVs recipient cells.

---

This evidence supports the unpredictable mechanism where SYNCRIP binding appears a prerequisite for PCBP2 recruitment. To further confirm and extend this observation, a number of mutants were designed and tested by RNA pull-down for the binding capacity of these two proteins; results indicate that (i) mutagenesis of the sole hEXO motif (miR-26b-3p no-hEXO) on miR-26 backbone (bearing both hEXO and CELL motifs), impairs also PCBP2 binding (*Figure 3A*) and conversely (ii) the de novo inclusion of a hEXO motif in miR-31-3p backbone (bearing only CELL-motifs) (miR-31-3p+hEXO) confers a de novo PCBP2 binding ability to this mutant; furthermore, mutation in the CELL motif (miR-31-3p+hEXO no-CELL) impairs PCBP2 binding (*Figure 3B*).

Overall, these data indicate that PCBP2 binding requires both the CELL motif and SYNCRIP binding; in other words, SYNCRIP binding is epistatic to PCBP2 recruitment.

## PCBP2 functionally dominates on SYNCRIP EV-loading activity on a repertoire of miRNAs embedding CELL and hEXO motifs

To extend the evidence for the role of PCBP2 in miRNA compartmentalization and to confirm its mechanistic role, (i) PCBP2 and (ii) SYNCRIP functional role in miRNA EVs/cell partition was evaluated, and (iii) PCBP2, (iv) SYNCRIP, and (v) SYNCRIP-dependent PCBP2 binding were assessed. First, NGS analysis of miRNAs exported in EVs produced by control and PCBP2-silenced murine hepatocytes allowed the selection of further miRNAs differentially loaded in EVs in correlation to PCBP2 (*Supplementary file 3*, *Figure 4A and B*).

Then, the functional role of PCBP2 was assessed by means of qRT-PCR performed on nine miRNAs expressed in EVs and embedding both CELL and hEXO motifs (see 'Materials and methods' section) in comparison to four miRNAs embedding either the CELL (miR-31-3p, miR-195a-5p, miR-421-3p) or the hEXO motif (miR-365-2-5p) (*Figure 4C*).

Specifically, miR-155-5p, miR-181d-5p, miR-3084-5p, miR-122b-3p, miR-192-5p, miR-26b-3p, miR-31-3p, miR-195a-5p, and miR-421-3p have the Core CELL motif (AUUA/G) described by Garcia-Martin and colleagues (*Garcia-Martin et al., 2022*). Other miRNAs (miR-345-3p, miR-23a-5p, and miR-214-3p) share the described CELL motif (*Garcia-Martin et al., 2022*) with the most frequent nucleotides, considering also the reported variability. Regarding the hEXO motif described by *Santangelo et al., 2016*, the most frequent nucleotides defining the motif sequence have been taken into consideration.

In order to consider the possible impact of PCBP2 on miRNAs steady state level (resulting from variation in transcription and biogenesis), miRNA abundance in EVs and in the intracellular compartment was analyzed by qRT-PCR as EVs/cell ratio. Results indicate that PCBP2 silencing releases a SYNCRIP-dependent loading of miRNAs in the EVs (*Figure 4D*), thus highlighting a dominant PCBP2 cell-retention function on SYNCRIP-dependent export, evaluated on these miRNAs in *Figure 4—figure supplement 1*.

Moreover, Gene Ontology (GO) enrichment analysis of the PCBP2-dependent cell-retained miRNAs targets have been performed (including both validated and predicted targets, respectively,

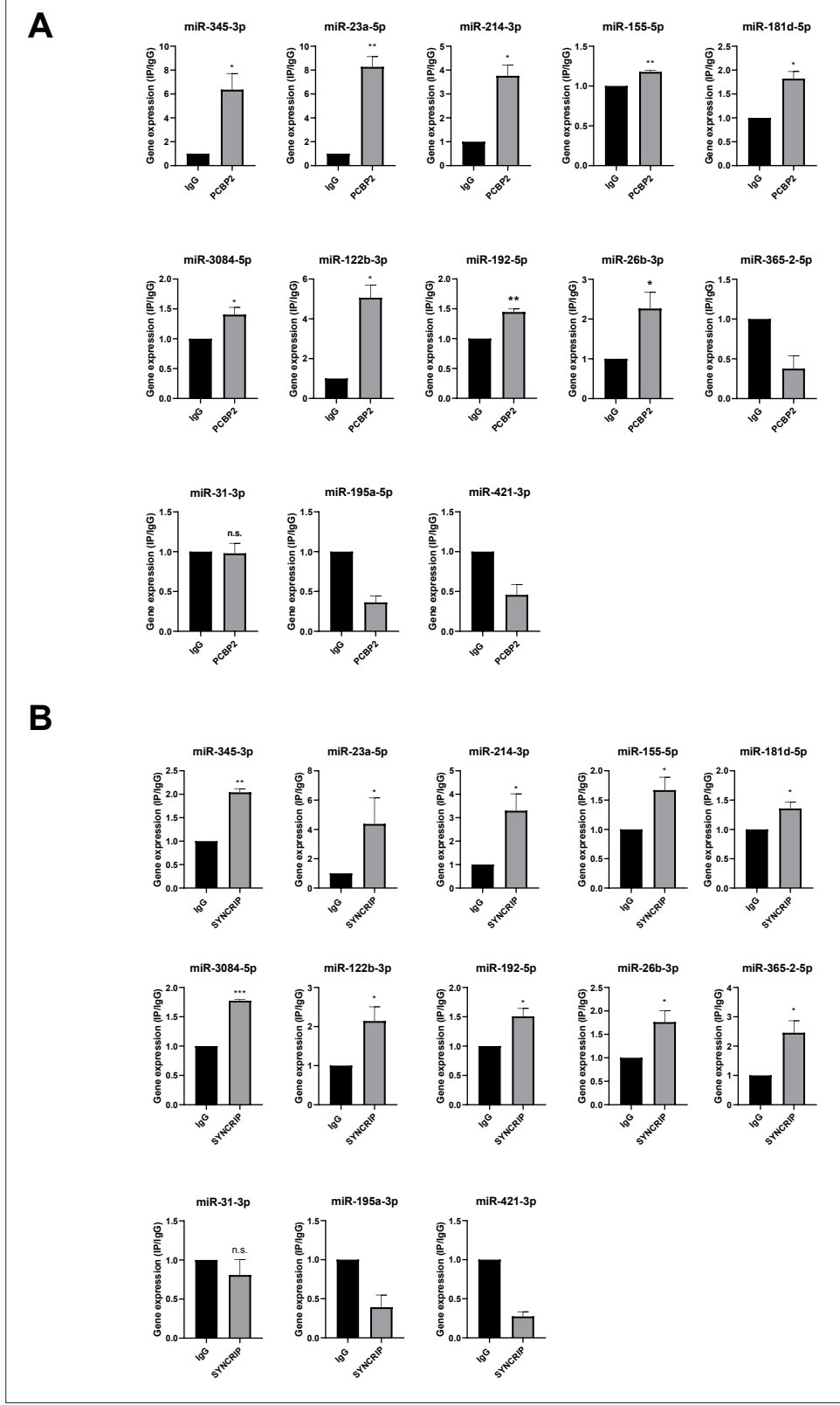

**Figure 5.** PCBP2 and SYNCRIP bind to several miRNAs embedding CELL and hEXO motif sequences. (**A**) CLIP of PCBP2 protein in murine hepatocytes. RT-qPCR analysis for the indicated miRNAs is shown as IP/IgG for each independent experiment (IgG arbitrary value 1). Data are the mean ± SEM of three independent experiments. Data are considered statistically significant with p<0.05 (Student's *t*-test). *p<0.05; **p<0.01. (**B**) CLIP of

*Figure 5 continued on next page*

*Figure 5 continued*

SYNCRIP protein in murine hepatocytes. RT-qPCR analysis for the indicated miRNAs is shown as IP/IgG for each independent experiment (IgG arbitrary value 1). Data are the mean ± SEM of three independent experiments. Data are considered statistically significant with p<0.05 (Student's *t*-test). *p<0.05; **p<0.01; ***p<0.001.

obtained from TarBase v9.0 database and DIANA-microT web server). As reported in *Figure 4E* and in *Figure 4—figure supplement 2*, this analysis highlighted several biological pathways collectively influenced by these miRNAs (e.g., establishment of organelle localization, regulation of cell cycle, and lymphocyte differentiation).

To preliminarily explore the functional impact of EVs deriving from WT and PCPB2-silenced cells, treatment of leukocytes obtained from C57BL/6J mice livers has been performed.

Specifically, as reported in *Figure 4F*, *Figure 4—figure supplement 3*, T cells, NK cells and myeloid cells show a higher amount of IFN-γ production upon *shPCBP2* EVs treatment in comparison to the *shCTR* EVs; this suggests that PCBP2 silencing results in an EV-mediated modulation of the immune response.

With respect to the molecular mechanism, the analysis by CLIP-qPCR assay demonstrates the PCBP2 binding to miRs-345-3p, 23a-5p, 214-3p, 155-5p, 181d-5p, 3084-5p, 122b-3p, 192-5p, 26b-3p

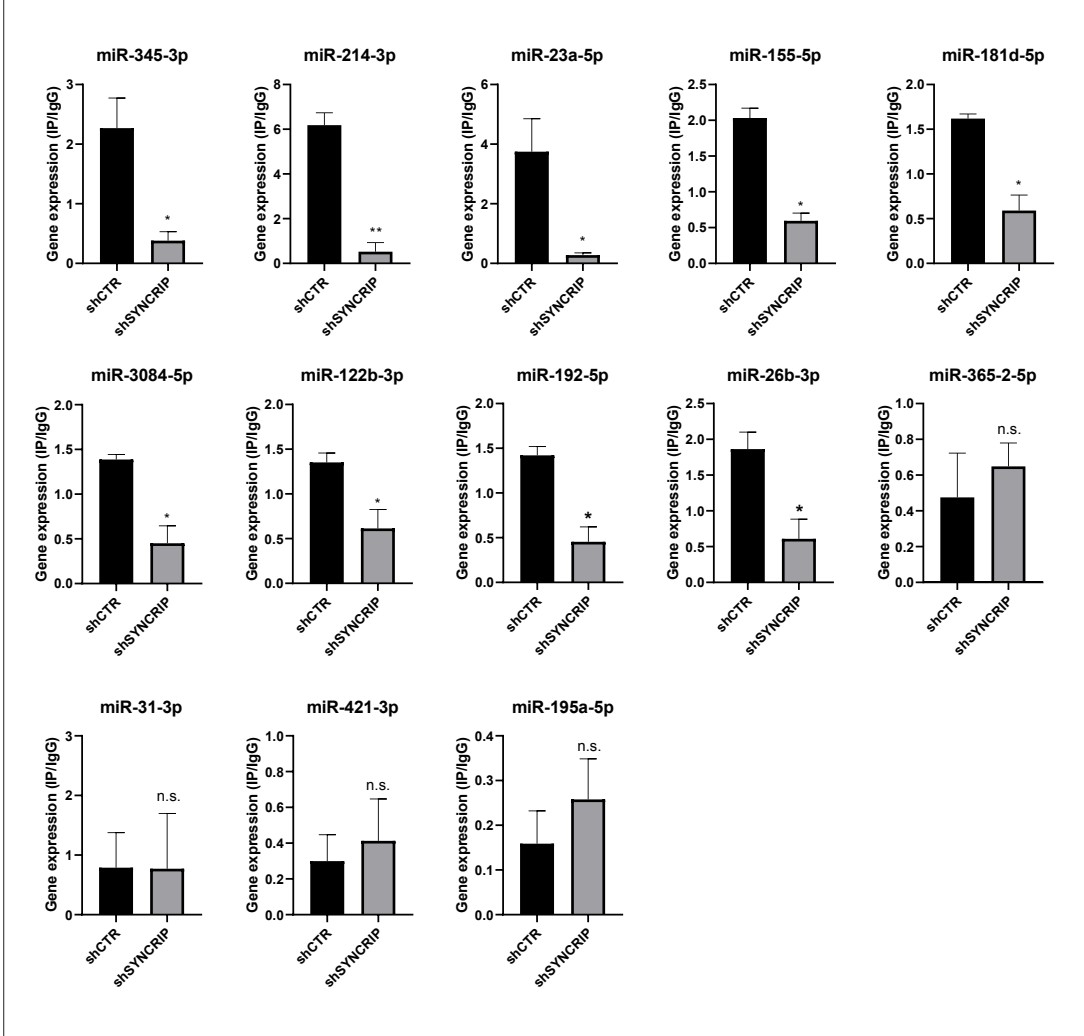

**Figure 6.** PCBP2 binding to miRNAs requires SYNCRIP. CLIP of PCBP2 protein in murine hepatocytes both WT (*shCTR*) and silenced for SYNCRIP (*shSYNCRIP*). RT-qPCR analysis for the indicated miRNAs is shown as IP/IgG. Data are the mean ± SEM of three independent experiments. Data are considered statistically significant with p<0.05 (Student's *t*-test). *p<0.05; ***p<0.001.

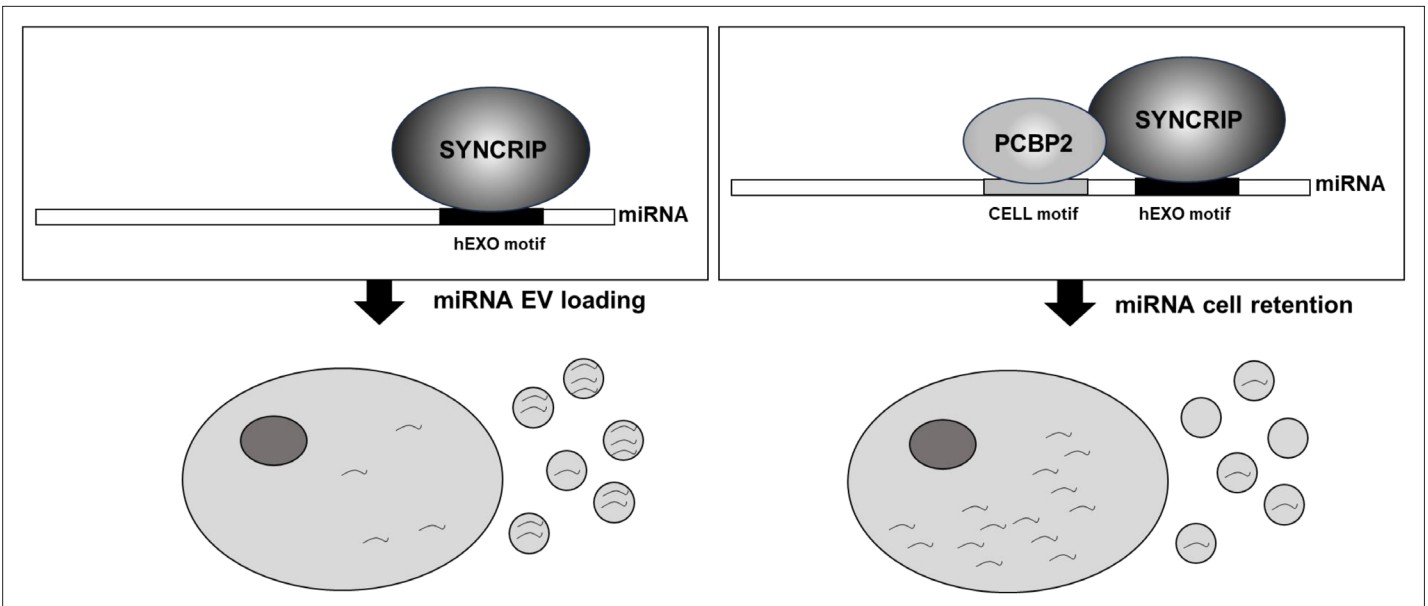

**Figure 7.** Schematic model of PCBP2/SYNCRIP-dependent miRNAs compartmentalization. hEXO-SYNCRIP interaction promotes miRNAs secretion into EVs. SYNCRIP-dependent PCBP2-CELL motif interaction promotes miRNAs intracellular retentio.

that bear both the CELL and hEXO motifs; conversely, the presence of either the sole hEXO (miR-365-2-5p) or the sole CELL (miR-31-3p, miR-195a-5p, miR-421-3p) does not allow PCBP2 binding (*Figure 5A*). As expected, the presence of hEXO motif alone or in combination with the CELL motif is sufficient for SYNCRIP binding to all the analyzed miRNAs (*Figure 5B*).

Furthermore, in line with data obtained for miR-155-3p (see *Figure 2D*), results shown in *Figure 6* demonstrate the SYNCRIP-dependent PCBP2 binding assessed in hepatocytes silenced for SYNCRIP.

Overall, these data indicate that the previously described SYNCRIP capacity to act as EV loader is functionally limited by the presence of a here-identified sequence- and SYNCRIP-dependent retention mechanism mediated by the RNA-binding protein PCBP2.

## Discussion

The main finding of this investigation is the identification of PCBP2 as a new regulator of miRNA partition between intracellular and EV compartments. Evidence here gathered indicates that (i) PCBP2 binding requires both the miRNA CELL sequence and the RBP SYNCRIP, which in turn recognizes its specific hEXO *consensus*; in other words, SYNCRIP binding is a prerequisite for PCBP2 recruitment; (ii) PCBP2 cell retention function is dominant over the SYNCRIP-dependent EV-loading; in other words, PCBP2 impairs SYNCRIP-mediated miRNA export (*Figure 7*).

While SYNCRIP EV-loading activity has been previously well characterized by means of both functional and structural analysis (*Santangelo et al., 2016*; *Hobor et al., 2018*), the role of PCBP2 as a mediator of miRNA intracellular retention is here disclosed for the first time.

Indeed, no data were previously reported on the RBPs involved in miRNA cell retention even if the role of specific *consensus* sequences has been previously defined by means of functional assays involving the introduction/removal of the CELL motifs (*Garcia-Martin et al., 2022*).

PCBP2 protein (similarly to SYNCRIP) displays a pleiotropic function; specifically, it is a well-characterized member of the Poly-rC-binding proteins (PCBPs), a group of multifunctional RNA-binding proteins that contain three highly conserved RNA binding KH domains and that may shuttle between the nucleus and the cytoplasm (*Yuan et al., 2021*). A large body of evidence points to its role in controlling multiple processes, including RNA maturation and trafficking, RNA editing, translational activation or repression, and mRNA degradation (*Chaudhury et al., 2010*; *Ishii et al., 2020*; *Makeyev et al., 1999*; *Makeyev and Liebhaber, 2002*; *Perrotti et al., 2002*; *Yabe-Wada et al., 2020*).

Its potential impact on miRNA biogenesis suggested us to analyze miRNA partition as the EVs/cell ratio, thus circumventing variation deriving from the intracellular expression levels.

Here, PCBP2 was found to directly bind to miRNAs, sharing one of the CELL-motifs previously identified by in silico sequence analysis of the miRNAs that were retained in AML12 mouse hepatocytes (*Garcia-Martin et al., 2022*). The use of specific insertion/removal mutants and knock-down cellular systems here highlighted the SYNCRIP recruitment as a prerequisite for PCBP2 binding, which was verified on miRs-155-3p, 26b-3p, and 31-3p. Furthermore, this conclusion was extended by CLIP analysis to further nine miRs selected on the basis of an NGS approach and of a bioinformatic analysis.

Of note, among them, miRs-155-3p, 23a-5p, 155-5p, 192-5p, and 26b-3p display important EV-mediated functions in relation to pathophysiology (*Broermann et al., 2020*; *Dosil et al., 2022*; *Jiao et al., 2021*; *Zhao et al., 2024*), as also assessed by GO analysis.

The here proposed mechanism implies that the export activity of SYNCRIP is specifically impaired by PCBP2 and highlights that the miRNA partition is not only related to the presence of specific RBPs/export sequences interaction. The final functional compartmentalization output appears the result of an integrated system of RNA/proteins interactions, here only partially unveiled, whose dynamics may provide elements for the explanation of the EVs miRNA cargo specificity and for its variation coherently to cellular plasticity.

In addition, it is conceivable to hypothesize that previously identified domains (*Hobor et al., 2018*; *Kim et al., 2000*; *Mizutani et al., 2000*; *Passos et al., 2006*; *You et al., 2009*), shall mediate the protein-protein interaction dynamics between PCBP2 and SYNCRIP, impacting RBP-miRNAs recognition. Moreover, upon the interaction among SYNCRIP and the miRNAs a topological switch may occur, impacting the affinity of PCBP2 for the same miRNAs.

Of note, recent research highlights a further level of complexity since miRNA epitranscriptomic modifications, while impairing miRNA intracellular function, appear instrumental to miRNA loading in EVs (*Garbo et al., 2024*).

The described multiple RNA/proteins interactions provide a further step in the process of clarification of the mechanisms that may yield value in the control of cellular communication in pathophysiological processes. Functionally, the effect of PCBP2 silenced cells-derived EVs on the immune response evaluated as IFN-γ production suggests modulation on functional states of recipient target cells. The knowledge of molecular players of miRNAs intracellular/EVs partition could be soon instrumental for the development of RNA-based manipulations holding therapeutic perspectives.

# Materials and methods
## Cell culture conditions

Non-tumorigenic murine hepatocyte 3A cells (*Montaldo et al., 2021*; *Conigliaro et al., 2013*) were grown at 37°C, in a humidified atmosphere with 5% $CO_2$, in RPMI 1640 medium supplemented with 10% FBS (Gibco Life Technology), 50 ng/mL epidermal growth factor (EGF), 30 ng/mL insulin growth

**Table 1.** Biotinylated RNA oligonucleotides used in pull-down experiments.

| Name | Oligonucleotides sequence |
| --- | --- |
| *Biotin-miR-26b-3p WT* | [Btn] 5' CCUGUUCUCCAUUACUUGGCUC 3' |
| *Biotin-miR-26b-3p no-hEXO* | [Btn] 5' CCUGUUCUCCAUUACUUGGCGA 3' |
| *Biotin-miR-31-3p WT* | [Btn] 5' UGCUAUGCCAACAUAUUGCCAUC 3' |
| *Biotin-miR-31-3p+hEXO* | [Btn] 5' UGCUAUGCCAACAUAUUGGGCUG 3' |
| *Biotin-miR-31–3p+hEXO no-CELL* | [Btn] 5' UGCUAUGCCAACAUAUUUGGCUG 3' |
| *Biotin-miR-155b-3p WT* | [Btn] 5' CUCCUACCUGUUAGCAUUAAC 3' |
| *Biotin-miR-155b-3p no-CELL* | [Btn] 5' CUCCUACCUGUUAGCAUGAUC 3' |
| *Biotin-miR-155b-3p no-hEXO* | [Btn] 5' CUCCUACCUGUUAGCAUUAGU 3' |
| *Biotin-miR-365-2-5p WT* | [Btn] 5' AGGGACUUUCAGGGGCAGCUGUG 3' |
| *Biotin-miR-365-2-5P+hEXO* | [Btn] 5' AGGGACUUUCAUUGGCAGCUGUG 3' |

factor (IGF) II (PeproTech), 10 mg/mL insulin (Roche), and penicillin/streptomycin, on dishes coated with collagen I (Collagen I, Rat Tail; Gibco Life Technology). All cell lines were tested for mycoplasma using the DAPI staining and the Mycoplasma PCR Detection Kit (G238, ABM). All cell lines were authenticated after thawing by morphology check, cell proliferation rate evaluation, and species verification by PCR. Bacteria contamination was excluded.

## Extracellular vesicle purification

EVs were prepared according to the International Society of Extracellular Vesicles recommendations (*Théry et al., 2018*). Conditioned media (CM) from 150 mm plates each containing 250,000 hepatocytes were collected after 72 h culture in complete medium containing EV-depleted FBS. Cell-conditioned media were centrifuged at 2000×$g$ for 20 min at 4°C to remove dead cells and then at 20,000×$g$ for 30 min at 4°C. Cleared supernatants were passed through 0.22 mm filter membranes, ultracentrifuged in a SW32 Ti rotor (Beckman Coulter) at 100,000×$g$ for 70 min at 4°C, and finally resuspended in PBS. The EVs resuspension was analyzed by EXOID-V1-SC (IZON) for size and concentration characterization.

## Biotin miRNA pull-down

Biotin miRNA pull-down experiments were performed on cytoplasmic extracts. Briefly, cells were lysed in hypotonic buffer (10 mM Tris-Cl [pH 7.5], 20 mM KCl, 1.5 mM MgCl$_2$, 5 mM DTT, 0.5 mM EGTA, 5% glycerol, 0.5% NP40, and 40 U/mL RNAsin [Promega]) supplemented with protease inhibitors (Roche Applied Science). Lysates were incubated on a rotating platform for 30 min at 4°C and then centrifuged at 13,000 rpm for 30 min at 4°C. Protein concentration was determined with Protein Assay Dye Reagent (Bio-Rad) based on the Bradford assay.

Samples (2 mg of proteins) were incubated for 1 h at 4°C with 10 nmol synthetic single-strand miRNA oligonucleotides containing a biotin modification attached to the 5' and via a spacer arm (IDT, Intregrated DNA Technology) (*Table 1*).

Dynabeads M-280 Streptavidin (50 μL/sample, Invitrogen), previously blocked with 1 mg/mL yeast tRNA (Roche Applied Science), were added to reaction mixture for 90 min at 4°C, and then the beads were washed three times with cold lysis buffer and once with PBS. Elution was performed at room temperature for 5 min in Laemmli Buffer (containing 2-β mercaptoethanol and SDS).

Detection of miRNA/RBPs interaction was evaluated by WB on 10% of Input sample and 50% of the pulled-down samples.

**Table 2.** Primers for miRNA qPCR analysis.

| miRNA | Primer sequence | $T_m$ (°C) |
|---|---|---|
| *mmu-miR-23a-5p* | GGGGTTCCTGGGGATGGGATTT | 60 |
| *mmu-miR-26b-3p* | CCTGTTCTCCATTACTTGGCTC | 62 |
| *mmu-miR-31-3p* | TGCTATGCCAACATATTGCCATC | 61 |
| *mmu-miR-122b-3p* | AAACACCATTGTCACACTCCAC | 60 |
| *mmu-miR-155-3p* | CTCCTACCTGTTAGCATTAAC | 59 |
| *mmu-miR-155-5p* | TTAATGCTAATTGTGATAGGGGT | 58 |
| *mmu-miR-181d-5p* | AACATTCATTGTTGTCGGTGGGT | 60 |
| *mmu-miR-192-5p* | CTGACCTATGAATTGACAGCC | 59 |
| *mmu-miR-195a-5p* | TAGCAGCACAGAAATATTGGC | 60 |
| *mmu-miR-214-3p* | ACAGCAGGCACAGACAGGCAGT | 60 |
| *mmu-miR-345-3p* | CCCTGAACTAGGGGTCTGGAGAC | 60 |
| *mmu-miR-365-2-5p* | GACTTTCAGGGGCAGCTG | 58 |
| *mmu-miR-3084-5p* | GTTGAAGGTTAATTAGCAGAGT | 60 |
| *mmu-miR-421-3p* | ATCAACAGACATTAATTGGGCGC | 60 |

**Table 3.** Primers for gene expression qPCR analysis.

| Name | Primer sequence | $T_m$ (°C) |
|---|---|---|
| PCBP2 | *For* ACACCGGATTCAGTGGCA<br>*Rev* TTGATTTTGGCGCCTTGACG | 58<br>58 |
| SYNCRIP | *For* ACCTTGCCAACACGTAACA<br>*Rev* CCATAGCCTTGACACACCA | 59<br>59 |
| 18s | *For* AGCACCCATTGCAACGTCTG<br>*Rev* GCACGGCGACTACCATCG | 58<br>58 |

Pull-down assay with PCBP2 (NM_001103165) mouse recombinant protein (TP522190, Origene) was performed with 4 µg of protein.

## Protein digestion, peptide purification, and nanoLC analysis

Proteins obtained from the pull-down experiments with miR-155-3p or random scrambled miRNA were separated on 4–12% gradient gels (Invitrogen) and stained by Simply Blue Safe Stain staining. Fourteen sections of the gel lane were cut. Protein digestion of gel pieces and peptide purification were performed as previously described in *Mancone et al., 2012*. Peptides resuspended in a suitable nanoLC injection volume of 2.5% ACN/0.1% TFA and 0.1% formic acid were then analyzed by an Ulti-Mate 3000 RSLCnano-LC system (Thermo Fisher Scientific) connected on-line via a nano-ESI source to an Q Exactive plus TM Hybrid Quadrupole-Orbitrap Mass Spectrometer (Thermo Fisher Scientific) as in *Montaldo et al., 2021*. Proteins were automatically identified by MaxQuant (v. 1.6.17.0) software. Tandem mass spectra were searched against the *Mus musculus* dataset of UniprotKB database.

Quantitative comparison among miR-155-3p WT and miR-155-3p no-CELL was performed using the label-free quantification algorithm calculated by MaxQuant software.

## SDS-PAGE and western blotting

Cells were lysed in Triton 1X buffer; subsequently, the proteins (30 µg for each sample) were analyzed as in *Battistelli et al., 2021*. The following primary antibodies were used for immunoblotting: α-PCBP2 (AV40568; Sigma-Aldrich), α- SYNCRIP (MAB11004; Merck Millipore), α-HSP90 (sc-13119; Santa Cruz Biotech), α-LAMP1 (Ab24170; Abcam), α-CD63 (sc-5275; Santa Cruz Biotechnology), α-SYNTHENIN (Ab133267; Abcam), α-ALIX (2171; Cell Signaling Technology), α-TSG101 (sc-7964; Santa Cruz Biotechnology), α-FLOTILLIN-1 (sc74566; Santa Cruz Biotechnology), α-CALNEXIN (NB100-1965; Novus Biologicals), and α-GAPDH (MAB-374; Merck Millipore) used as a loading control. The immune complexes were detected with horseradish peroxidase-conjugated species-specific secondary anti-serum: (α-Rabbit 172-1019 and α-Mouse 170-6516; Bio-Rad Laboratories), then by enhanced chemi-luminescence reaction (Bio-Rad Laboratories). Densitometric analysis of protein expression was performed by using the Fiji-ImageJ image processing package.

## RNA extraction, RT-PCR, and real-time qPCR

miRNAs were extracted by miRNeasy Mini Kit and RNeasy MinElute Cleanup Kit (QIAGEN) and reverse transcribed with MystiCq microRNA cDNA Synthesis Mix (Sigma-Aldrich). Quantitative polymerase chain reaction (RT-qPCR) analyses were performed according to MIQE guidelines. cDNAs were ampli-fied by qPCR reaction using GoTaq qPCR Master Mix (Promega, Madison, WI, USA). Relative amounts, obtained with $2^{(-\Delta Ct)}$ method, were normalized with respect to the cel-miR-39 Spike-In (59000; NORGEN), previously added into miRNA samples and expressed as ratio of miRNAs expression in EVs with respect to the intracellular compartment *Garbo et al., 2024*. Oligonucleotide sequences are reported in *Table 2*.

Total RNA was extracted by ReliaPrep RNA Tissue Miniprep System (Promega, Madison, WI, USA) and reverse transcribed with iScript c-DNA Synthesis Kit (Bio-Rad Laboratories Inc, USA). Quantitative polymerase chain reaction (RT-qPCR) analyses were performed according to MIQE guidelines. cDNAs were amplified by qPCR reaction using GoTaq qPCR Master Mix (Promega, Madison, WI, USA). Rela-tive amounts, obtained with $2^{(-\Delta Ct)}$ method, were normalized with respect to the housekeeping gene 18S. Oligonucleotide sequences are reported in *Table 3*.

The results were analyzed with Manager Software (Bio-Rad) and calculated using the ΔC(t) method.

**Table 4.** Oligos for shRNA cloning in pSUPER.retro.puro vector.

| Name | Sequence |
|------|----------|
| PCBP2 | *Sense* GATCCCCGAGCAGACCCATCCA TAATTTCAAGAGAATTATGGATGGGTCTGCTCTTTTTA *Antisense* AGCTTAAAAAGAGCAGACCCATCCATA ATTCTCTTGAAATTATGGATGGGTCTGCTCGGG |
| SYNCRIP | As reported in *Santangelo et al., 2016* |
| CTR | As reported in *Santangelo et al., 2016* |

## Co-immunoprecipitation

Cells were lysed with IP Lysis Buffer (150 mM NaCl, 50 mM Tris-HCl pH 7.5, 5 mM EGTA pH 8, 50 mM NaF pH 8, 1,5 mM MgCl$_2$, 1% TRITON-X100, and 10% glycerol) containing freshly added cocktail protease inhibitors (complete EDTA-free Protease Inhibitor Cocktail; Sigma-Aldrich) and phosphatase inhibitors (5 mM EGTA pH 8.0; 50 mM sodium fluoride; 5 mM sodium orthovanadate). Lysates were incubated on a rotating platform for 2 h at 4°C and then centrifuged at 13,000 rpm for 30 min at 4°C. Protein concentration was determined with Protein Assay Dye Reagent (Bio-Rad), based on the Bradford assay.

2 mg of proteins (one for the specific antibody and one for the corresponding aspecific IgG) were precleared adding 40 µL of Protein A Sepharose or Protein G Sepharose (GE HealthCare) for 3 hr at 4°C in a total volume of 1 mL of IP Lysis Buffer in rotation. Then, Protein A or G Sepharose was removed by centrifugation and the extracts were incubated with 5 µg of specific antibody α-PCBP2 (cod. RN025P; MBL), SYNCRIP (MAB11004; Merck Millipore), Normal Rabbit IgG (12-370; Millipore), or Normal Mouse IgG (12-371; Merck Millipore), the last two used as negative controls, to proceed with immunoprecipitation at 4°C overnight. Immuno-complexes were collected adding 50 µL of Protein A or G Sepharose for 3 h at 4°C in rotation. The immunoprecipitated proteins were washed three times with Net Gel Buffer (150 mM NaCl, 50 mM Tris-HCl pH 7.5, 1 mM EDTA, 0.1% NP40, and 0.25% gelatin) and once with RIPA Buffer (150 mM NaCl, 1% NP40, 0,5% sodium deoxycholate, 0,1% SDS, 50 mM Tris-HCl pH 8). Finally, immunoprecipitated proteins were detached from Sepharose beads by adding 50 µL of Laemmli Buffer 2X. Samples were boiled at 95°C for 5 min, beads were eliminated by centrifugation, and 10% of input sample and 50% of each immunoprecipitated sample were loaded on polyacrylamide gel and analyzed by western blotting.

## EMSA

Cells were lysed in Triton Buffer at 4°C, for 30 min, and 4 µg of protein extract were incubated with 0.5 pmol of biotinylated RNA oligonucleotides for 30 min at room temperature in REMSA Binding Buffer, according to the manufacturer's protocol (Light Shift Chemiluminescent RNA EMSA Kit, Thermo Fisher Scientific 20158). 1 µg of each antibody was incubated with the protein-RNA complex: anti-PCBP2 (RN025P; MBL) and anti-SYNCRIP (MAB11004; Merck Millipore) for supershift and ultrashift analysis. The electrophoresis was performed in native 6% polyacrylamide gel in 0.5× TBE. Transfer step was carried out at 25 V, for 15 min in 0.5× TBE, and the detection was performed following manufacturer's instructions.

## UV cross-linking RIP

CLIP was performed as reported in *Battistelli et al., 2021*. Immunoprecipitated miRNAs were reverse transcribed and analyzed by RT-qPCR amplifications. The list of primers is reported in *Table 2*. Primary antibodies for IP: anti-PCBP2 (RN025P; MBL), anti-SYNCRIP (MAB11004; Merck Millipore) and as negative controls Normal Rabbit IgG (12-370; Merck Millipore) or Normal Mouse IgG (12-371; Merck Millipore).

## shRNA silencing

Stable PCBP2 knockdown was achieved through infection with shRNAs cloned in pSUPER retro puro retroviral vector (Oligoengine). Viral supernatants were collected 48 h after transfection of 293gp packaging cells, filtered (0.45 mm), and added to hepatocytes. At 48 h post-infection, selection was

performed with 2 µg/mL puromycin for at least 1 week before analysis. The sequence of shRNA scramble used as control was previously described (*Pasque et al., 2011*). The sequences of shRNA oligos used for cloning are reported in *Table 4*.

## Motif scanning analysis

Murine mature miRNA sequences were retrieved from miRBase v22.1 database (*Kozomara et al., 2019*). The FIMO tool (*Grant et al., 2011*) was used to scan these sequences for occurrences of hEXO, extended CELL (the bottom motif identified in AML12 cells and reported in *Figure 2* by *Garcia-Martin et al., 2022*) and core AUUA/G CELL motifs, encoded as Position Probability Matrices, with parameters `--bfile --motif-- --norc` and setting the p-value threshold to 0.1, 0.1, and 0.01, respectively.

## Small RNA sequencing

miRNA samples (two biological replicates per condition), to which the cel-miR-39 Spike-In (59000; NORGEN) was previously added, were sequenced at Procomcure Biotech GmbH. Sequencing libraries were prepared using the NEXTFLEX Small RNA-Seq Kit v4 (PerkinElmer). The sequencing reaction was performed on an Illumina NovaSeq 6000 instrument in 2 × 40 bp paired-end configuration, with a throughput of ~40 million read pairs per sample. FastqToolkit version 2.2.5 (*BaseSpace Labs, 2023*) was used to remove adapter sequences from the 3' end and to filter out reads whose length and average quality after trimming were <10 and <30, respectively. Only forward reads were kept for downstream analyses. The mirPRo software version 1.1.4 (*Shi et al., 2015*), which utilizes NovoAlign (*Li and Homer, 2010*) as its alignment engine, was used to align reads to a reference composed of miRNA hairpin sequences downloaded from the miRBase v22.1 database with the addition of the spike-in and to count reads mapping to mature miRNAs. A count matrix was assembled, including only mature miRNAs with one or more reads in at least two cell and two EV samples. Differential abundance analysis was performed using the DESeq2 R package (*Love et al., 2014*). Size factors were estimated directly from spike-in counts. For each mature miRNA, a likelihood ratio test was conducted to assess differences between the EV/cell abundance ratios measured in the *shPCBP2* and *shCTR* conditions.

## miRNA target prediction and enrichment analysis

Enrichment analysis was performed using the R Bioconductor packages ClusterProfiler (v4.14.4), org. Mm.eg.db (v3.20.0), and AnnotationDbi (v1.68.0). The analysis was carried out on miRNA validated targets identified through TarBase v9.0 database (https://dianalab.e-ce.uth.gr/tarbasev9) and miRNA targets predicted with DIANA-MicroT Webserver (https://dianalab.e-ce.uth.gr/microt_webserver/#/), using miRBase v22.1 as miRNA resource. GO terms for Biological Processes were filtered using p-value and q-value cutoffs, respectively, of 0.05 and 0.2. Multiple testing correction was applied using the Benjamini–Hochberg method. Enriched GO terms were visualized using dotplots.

## Ex vivo analysis

Female C57BL/6J mice were purchased from Charles River and were of 12–14 weeks old by the time of use weighing 20–25 g. All animals were housed in ventilated cages (no more than five mice per cage) under specific pathogen-free conditions and in a controlled environment (12 h daylight cycle, lights off at 18:00) with free access to food and water. Livers were dissected from mice, smashed using PBS, and strained (100 µm filter). Leukocytes, enriched using 40% Percoll, were plated at a concentration of 1 × 10$^6$/mL in complete RPMI, and supplemented with EVs-free 10% fetal calf serum, 100 U/mL penicillin/streptomycin, 1% glutamine, and with 20 ng/mL recombinant murine IL-2 (Peprotech) and 20 ng/mL recombinant murine IL-7 (Peprotech) at 37°C under 5% $CO_2$. Leukocytes were cultured either alone or in the presence of 10 µg/mL control EVs or with the same amount of *shPCBP2* EVs for 24 h at 37°C under 5% $CO_2$. Brefeldin A 5 µg/mL (Merck Life Sciences) was added in the last 3 h of treatment.

Cells were stained for cell surface markers using the following antibodies: anti-CD45.2-PercP-Cy5.5 (clone 194, Invitrogen); anti-NK1.1-PE (clone PK136, Invitrogen); anti-CD11b-BV421 (clone M1/70, BioLegend); anti-CD3-BV510 (clone 145-2C11, BioLegend); and APC-H7-conjugated Fixable viability Dye (Thermo Scientific) to identify dead cells. Cells were then fixed and permeabilized using the Cytofix/cytoperm kit (BD Biosciences) before intracellular staining with anti-IFN-γ-APC (clone XMG1.2,

BioLegend). Samples were acquired by FacsCantoII and Flow cytometric analysis was performed with FlowJo 10 software (BD Bioscience). Live/dead(L/D)-/CD45.2+ cells were gated and then segregated into NK1.1+ CD11b+, NK1.1- CD11b+, NK1.1- CD11b- CD3+ cells to discriminate NK cells, myeloid cells, and T cells, respectively. These gated cells were then analyzed for the expression of IFN-γ as reported in *Figure 4—figure supplement 3*.

Animal studies were conducted in accordance with all relevant ethical regulations for animal testing and research including the Italian code for the care and use of animals for scientific purposes.

The Italian Ministry of Health approved the use of animals (authorization no. 698/2021-PR).

## Statistical analysis

For the qRT-PCR analysis, and for ex vivo analysis, statistical differences were assessed with the one-tailed paired Student's *t*-test using GraphPad Prism version 9 (GraphPad Software). Data are presented as mean ± SEM, and p-values<0.05 were considered statistically significant. For the statistical analysis of proteomic studies, Perseus software (version 1.6.7.0) after log2 transformation of the intensity data was used. Results were considered statistically significant at p<0.05.

## Materials availability

This study generated a new cell line (3A *shPCBP2*) and used previously generated cell lines (3A *shCTR* and 3A *shSYNCRIP*); we are available to share them under request without restrictions.

## Acknowledgements

We thank Prof. Andres Ramos and Prof. Rosa Molfetta for suggestions and critical reading of the manuscript, Andrea Melito, Cristina Mordenti, and Claudia Maldonado Torres for their help in molecular analyses and data collection. SG is supported by the AIRC Post-Doctoral Fellowship. This work was funded by Associazione Italiana per la Ricerca sul Cancro (IG26290) to MT, (IG24955) to RP, and by the Italian Ministry of Health 'Ricerca Corrente-Linea 2-Progetto 5-INMI L.Spallanzani IRCCS' funding to CM; Istituto Pasteur-Fondazione Cenci-Bolognetti (Anna Tramontano grant 2020) to CC; European Research Council (RIBOMYLOME_309545 and ASTRA_855923) and the H2020 projects (IASIS_727658 and INFORE_825080) to GGT; Sapienza University of Rome (RM12218166AEFC72) to CB and (MA12117A5D73CC9C) to MT; SEED PNR-Finanziamento di progetti di ricerca su temi di interesse trasversale per il PNR 2021 to CB; PNRR Sapienza-Rome Technopole per l'internazionalizzazione della ricerca, project NanoSeAl to CB; PNRR-Rome Technopole to MT and PRIN: progetti di ricerca di rilevante interesse nazionale–Bando 2022 Prot. 2022ETPX42 to MT.

## Additional information

### Funding

| Funder | Grant reference number | Author |
|---|---|---|
| Associazione Italiana per la Ricerca sul Cancro | IG26290 | Marco Tripodi |
| Associazione Italiana per la Ricerca sul Cancro | IG24955 | Rossella Paolini |
| Ricerca Corrente-Linea 2-Progetto 5-INMI L.Spallanzani IRCCS | | Claudia Montaldo |
| Istituto Pasteur-Fondazione Cenci Bolognetti | 2020 | Carla Cicchini |
| European Research Council | RIBOMYLOME_309545 | Gian Gaetano Tartaglia |
| European Research Council | 10.3030/855923 | Gian Gaetano Tartaglia |

| Funder | Grant reference number | Author |
| --- | --- | --- |
| H2020 European Research Council | 10.3030/727658 | Gian Gaetano Tartaglia |
| H2020 European Research Council | INFORE_825080 | Gian Gaetano Tartaglia |
| Sapienza University of Rome | RM12218166AEFC72 | Cecilia Battistelli |
| Sapienza University of Rome | MA12117A5D73CC9C | Marco Tripodi |
| SEED PNR-Finanziamento di progetti di ricerca su temi di interesse trasversale per il | PNR 2021 | Cecilia Battistelli |
| PNRR Sapienza-Rome Technopole per l'internazionalizzazione della ricerca, project NanoSeAl | | Cecilia Battistelli |
| PNRR-Rome Technopole | | Marco Tripodi |
| Ministero dell'Università e della Ricerca | PRIN: progetti di ricerca di rilevante interesse nazionale–Bando 2022ETPX42 | Marco Tripodi |

The funders had no role in study design, data collection and interpretation, or the decision to submit the work for publication.

## Author contributions

Francesco Marocco, Sabrina Garbo, Data curation, Formal analysis, Validation, Investigation, Methodology, Writing – review and editing; Claudia Montaldo, Mario Lecce, Alessia Carnevale, Data curation, Formal analysis, Methodology; Alessio Colantoni, Data curation, Software, Formal analysis, Methodology, Writing – review and editing; Luca Quattrocchi, Data curation, Software, Investigation, Methodology; Gioele Gaboardi, Formal analysis, Validation, Methodology; Giovanna Sabarese, Formal analysis; Carla Cicchini, Resources, Supervision, Funding acquisition; Rossella Paolini, Resources, Data curation, Supervision, Funding acquisition, Investigation, Writing – review and editing; Gian Gaetano Tartaglia, Resources, Data curation, Software, Supervision, Funding acquisition, Investigation, Methodology; Cecilia Battistelli, Marco Tripodi, Conceptualization, Resources, Supervision, Funding acquisition, Investigation, Writing – original draft, Project administration, Writing – review and editing

## Author ORCIDs

Francesco Marocco ⓘ https://orcid.org/0000-0002-4836-057X
Sabrina Garbo ⓘ https://orcid.org/0000-0001-7192-0013
Giovanna Sabarese ⓘ https://orcid.org/0000-0002-6841-1610
Cecilia Battistelli ⓘ https://orcid.org/0000-0002-5611-7377

## Ethics

Animal studies were conducted in accordance with all relevant ethical regulations for animal testing and research including the Italian code for the care and use of animals for scientific purposes. The Italian Ministry of Health approved the use of animals (Authorization n. 698/2021-PR).

Reviewer #1 (Public review): https://doi.org/10.7554/eLife.105017.4.sa1
Reviewer #2 (Public review): https://doi.org/10.7554/eLife.105017.4.sa2
Author response https://doi.org/10.7554/eLife.105017.4.sa3

# Additional files

## Supplementary files
Supplementary file 1. Results of proteomic analysis of proteins bound to miRNA-155-3p no-CELL vs WT.

Supplementary file 2. Results of proteomic analysis of proteins differentially bound to miRNA-155-3p no-CELL vs WT.

Supplementary file 3. Results of the likelihood ratio test performed on small RNA sequencing data to assess the differential miRNA EV loading between *shPCBP2* and *shCTR* conditions.

MDAR checklist

## Data availability
The miRNA-seq data generated in this study have been deposited and are available in the GEO database under accession code GSE269709.

The following dataset was generated:

| Author(s) | Year | Dataset title | Dataset URL | Database and Identifier |
|---|---|---|---|---|
| Marocco F, Garbo S, Montaldo C, Gaboardi G, Colantoni A, Cicchini C, Tartaglia GG, Ramos A, Battistelli C, Tripodi M | 2025 | The RNA-binding protein PCBP2 is a regulator of microRNAs partition between cell and extracellular vesicles | https://www.ncbi.nlm.nih.gov/geo/query/acc.cgi?acc=GSE269709 | NCBI Gene Expression Omnibus, GSE269709 |

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
