## [Editor Report · eLife Assessment]

This article makes a **valuable** contribution to the field by uncovering a molecular mechanism for miRNA intracellular retention, mediated by the interaction of PCBP2, SYNCRIP, and specific miRNA motifs. The findings are **convincing** and advance our understanding of RNA-binding protein-mediated miRNA sorting, providing deeper insights into miRNA dynamics.

---

## [Referee Report · Reviewer #1 (Public review)]

In this study, Marocco and colleagues perform a deep characterization of the complex molecular mechanism guiding the recognition of a particular CELLmotif previously identified in hepatocytes in another publication. Having miR-155-3p with or without this CELLmotif as initial focus, the authors identify 21 proteins differentially binding to these two miRNA versions. From these, they decided to focus on PCBP2. They elegantly demonstrate PCBP2 binding to miR-155-3p WT version but not to the CELLmotif-mutated version. miR-155-3p contains a hEXOmotif identified in a different report, whose recognition is largely mediated by another RNA-binding protein called SYNCRIP. Interestingly, mutation of the hEXOmotif contained in miR-155-3p did not only blunt SYNCRIP binding, but also PCBP2 binding despite the maintenance of the CELLmotif. This indicates that somehow SYNCRIP binding is a prerequisite for PCBP2 binding. EMSA assay confirms that SYNCRIP is necessary for PCBP2 binding to miR-155-3p, while PCBP2 is not needed for SYNCRIP binding. The authors aim to extend these findings to other miRNAs containing both motifs. For that, they perform a small-RNA-Seq of EVs released from cells knockdown for PCBP2 versus control cells, identifying a subset of miRNAs whose expression either increases or decreases. The assumption is that those miRNAs containing PCBP2-binding CELLmotif should now be less retained in the cell and go more to extracellular vesicles, thus reflecting a higher EV expression. The specific subset of miRNAs having both the CELLmotif and hEXOmotif (9 miRNAs) whose expressions increase in EVs due to PCBP2 reduction is also affected by knocking down SYNCRIP in the sense that reduction of SYNCRIP leads to lower EV sorting. Further experiments confirm that PCBP2 and SYNCRIP bind to these 9 miRNAs and that knocking down SYNCRIP impairs their EV sorting.

---

## [Referee Report · Reviewer #2 (Public review)]

Summary:

The author of this manuscript aimed to uncover the mechanisms behind miRNA retention within cells. They identified PCBP2 as a crucial factor in this process, revealing a novel role for RNA-binding proteins. Additionally, the study discovered that SYNCRIP is essential for PCBP2's function, demonstrating the cooperative interaction between these two proteins. This research not only sheds light on the intricate dynamics of miRNA retention but also emphasizes the importance of protein interactions in regulating miRNA behavior within cells.

Strengths:

This paper makes important progress in understanding how miRNAs are kept inside cells. It identifies PCBP2 as a key player in this process, showing a new role for proteins that bind RNA. The study also finds that SYNCRIP is needed for PCBP2 to work, highlighting how these proteins work together. These discoveries not only improve our knowledge of miRNA behavior but also suggest new ways to develop treatments by controlling miRNA locations to influence cell communication in diseases. The use of liver cell models and thorough experiments ensures the results are reliable and show their potential for RNA-based therapies.

---

## [Author Response]

The following is the authors’ response to the previous reviews

**Public Reviews:**

**Reviewer #1 (Public review):**
In this study, Marocco and colleages perform a deep characterization of the complex molecular mechanism guiding the recognition of a particular CELLmotif previously identified in hepatocytes in another publication. Having miR-155-3p with or without this CELLmotif as initial focus, authors identify 21 proteins differentially binding to these two miRNA versions. From these, they decided to focus on PCBP2. They elegantly demonstrate PCBP2 binding to miR-155-3p WT version but not to CELLmotif-mutated version. miR-155-3p contains a hEXOmotif identified in a different report, whose recognition is largely mediated by another RNA-binding protein called SYNCRIP. Interestingly, mutation of the hEXOmotif contained in miR-155-3p did not only blunt SYNCRIP binding, but also PCBP2 binding despite the maintenance of the CELLmotif. This indicates that somehow SYNCRIP binding is a pre-requisite for PCBP2 binding. EMSA assay confirms that SYNCRIP is necessary for PCBP2 binding to miR-155-3p, while PCBP2 is not needed for SYNCRIP binding. Then authors aim to extend these finding to other miRNAs containing both motifs. For that, they perform a small-RNA-Seq of EVs released from cells knockdown for PCBP2 versus control cells, identifying a subset of miRNAs whose expression either increases or decreases. The assumption is that those miRNAs containing PCBP2-binding CELLmotif should now be less retained in the cell and go more to extracellular vesicles, thus reflecting a higher EV expression. The specific subset of miRNAs having both the CELLmotif and hEXOmotif (9 miRNAs) whose expressions increase in EVs due to PCBP2 reduction is also affected by knocking-down SYNCRIP in the sense that reduction of SYNCRIP leads to lower EV sorting. Further experiments confirm that PCBP2 and SYNCRIP bind to these 9 miRNAs and that knocking down SYNCRIP impairs their EV sorting.In the revised manuscript, the authors have addressed most of my concerns and questions. I believe the new experiments provide stronger support for their claims. My only remaining concern is the lack of clarity in the replicates for the EMSA experiment. The one shown in the manuscript is clear; however, the other three replicates hardly show that knocking down SYNCRIP has an effect on PCBP2 binding. Even worse is the fact that these replicates do not support at all that PCBP2 silencing has no effect on SYNCRIP binding, as the bands for those types of samples are, in most of the cases, not visible. I think the authors should work on repeating a couple of times EMSA experiment.

We thank this Reviewer for having appreciated the novelty and the robustness of our data. In accordance with the Reviewer’s concern, we repeated the EMSA assay, specifically to address the PCBP2-independent SYNCRIP binding. In Author response image 1, we report the new EMSA replicates (top), the quantification of each signal (bottom) and the mean of EMSA signals relative to the three independent experiments (right). We hope that the new evidence will meet the required standards.

**Reviewer #2 (Public review):**
Summary:The author of this manuscript aimed to uncover the mechanisms behind miRNA retention within cells. They identified PCBP2 as a crucial factor in this process, revealing a novel role for RNAbinding proteins. Additionally, the study discovered that SYNCRIP is essential for PCBP2's function, demonstrating the cooperative interaction between these two proteins. This research not only sheds light on the intricate dynamics of miRNA retention but also emphasizes the importance of protein interactions in regulating miRNA behavior within cells.Strengths:This paper makes important progress in understanding how miRNAs are kept inside cells. It identifies PCBP2 as a key player in this process, showing a new role for proteins that bind RNA. The study also finds that SYNCRIP is needed for PCBP2 to work, highlighting how these proteins work together. These discoveries not only improve our knowledge of miRNA behavior but also suggest new ways to develop treatments by controlling miRNA locations to influence cell communication in diseases. The use of liver cell models and thorough experiments ensures the results are reliable and show their potential for RNA-based therapiesWeaknesses:The manuscript is well-structured and presents compelling data, but I noticed a few minor corrections that could further enhance its clarity:Figure References: In the response to Reviewer 1, the comment states, "It's not Panel C, it's Panel A of Figure 1"-this should be cross-checked for consistency.Supplementary Figure 2 is labeled as "Panel A"-please verify if additional panels (B, C, etc.) are intended.Western Blot Quality: The Alix WB shows some background noise. A repeat with optimized conditions (or inclusion of a cleaner replicate) would strengthen the data. Adding statistical analysis for all WBs would also reinforce robustness.These are relatively small refinements, and the manuscript is already in excellent shape. With these adjustments, it will be even stronger.

We deeply thank this Reviewer for having considered this new version of the manuscript and for having described its shape as excellent. In order to address the Reviewer’s concerns, we crosschecked the consistency of the described figures’ panels described in the text accordingly. Regarding the qualitative analysis of EV markers, we repeated the western blot analysis with optimized conditions as suggested and included the new panel (Author response image 2) in the supplementary figure 2, allowing to appreciate the signal relative to ALIX expression.

**Author response image 2. sa3fig2:** 

**Recommendations for the authors:**

**Reviewer #2 (Recommendations for the authors):**
Careful reading is required to rectify typo errors.

We thank the Reviewer for this suggestion. We amended the text to rectify typo errors.